# Production and Characterization of Porous Polymeric Membranes of PLA/PCL Blends with the Addition of Hydroxyapatite

**Nayara Koba de Moura [1], Idália A. W. B. Siqueira [1], João Paulo de Barros Machado [2], Hueliton Wilian Kido [3], Ingrid Regina Avanzi [3], Ana Claudia Muniz Rennó [3], Eliandra de Sousa Trichês [1] and Fabio Roberto Passador [1,*]**

[1] Polymer and Biopolymer Technology Laboratory (TecPBio), Department of Science and Technology, Federal University of São Paulo (UNIFESP), 330 Talim St, São José dos Campos 12231-280, SP, Brazil; nayara.koba.de.moura@gmail.com (N.K.d.M.); idalia_siqueira@yahoo.com.br (I.A.W.B.S.); eliandra.sousa@gmail.com (E.d.S.T.)

[2] National Institute for Space Research (INPE), 1758 dos Astronautas Avenue, São José dos Campos 12227-010, SP, Brazil; machadopaulo@gmail.com

[3] Department of Bioscience, Federal University of São Paulo (UNIFESP),136 Silva Jardim St, Santos 11015-020, SP, Brazil; kidohw@gmail.com (H.W.K.); avanzi@unifesp.br (I.R.A.); a.renno@unifesp.br (A.C.M.R.)

\* Correspondence: fabio.passador@unifesp.br; Tel.: +55-12-3924-9500 (ext. 9738)

**Abstract:** Polymer membranes have been widely used in guided tissue regeneration (GTR) and guided bone regeneration (GBR). The literature recognizes that poly (lactic acid) (PLA)/poly ($\varepsilon$-caprolactone) (PCL) blends have better physicochemical properties and that a porous polymer surface facilitates cell adhesion and proliferation. In addition, hydroxyapatite (HAp) incorporated into the polymer matrix promotes osteoinduction properties and osteoconduction to the polymer-ceramic biocomposite. Therefore, polymer membranes of PLA/PCL blend with the addition of HAp could be an alternative to be used in GBR. HAp was obtained by precipitation using the mixture of solutions of tetrahydrate calcium nitrate and monobasic ammonium phosphate salts. The porous membranes of the PLA/PCL (80/20) blend with the addition of HAp were obtained by solvent casting with a controlled humidity method, with the dispersion of HAp in chloroform and subsequent solubilization with the components of the blend. The solution was poured into molds for solvent evaporation under a controlled humidity atmosphere. The membranes showed the formation of pores on their surface, together with dispersed HAp particles. The results showed an increase in the surface porosity and improved bioactivity properties with the addition of HAp. Moreover, in biological studies with cell culture, it was possible to observe that the membranes with HAp have no cytotoxic effect on MC3T3 cells. These results indicate a promising use of the new biomaterial for GBR.

**Keywords:** porous polymeric membranes; PLA; PCL; hydroxyapatite; GBR

## 1. Introduction

Guided bone regeneration (GBR) is a technique that seeks for the regeneration of bone defects using a physical barrier that prevents the approach of non-osteogenic cells into the bone wound [1]. Generally, the membranes used in GBR are roughly divided into two types: bioabsorbable and non-resorbable membranes [2]. The bioabsorbable membranes can be sorted into natural polymers, synthetic polymer materials, and/or polymer composites that are a combination of two or more different materials used to obtain specific mechanical, chemical, and physical properties. Among the synthetic polymers, aliphatic polymers are excellent candidates for the production of membranes for this purpose. These polymers

allow for the production of stable porous materials that do not dissolve or fuse in tissue culture in vitro and are suitable for fabrication of three-dimensional scaffolds [3]. The major polymers with these characteristics include poly (lactic acid) (PLA) and polycaprolactone (PCL) [1,4]. The membranes prepared with these individual polymers have excellent biocompatibility, controllable biodegradability, low rigidity, easy processability, and drug-encapsulating ability [4,5].

PLA has good processability, mechanical resistance, and its degradation can produce oligomers and monomers of lactic acid, which are completely absorbable by the organism. The use of PLA for application in GBR is interesting due to its thermal properties. The crystalline state of PLA can vary from completely amorphous (non-crystalline) to up to 40% crystalline. The PLA has a glass transition temperature (Tg) that ranges from 50 to 80 °C and a melting temperature (Tm) that ranges from 130 to 180 °C [6–8].

PCL has a hydrophobic character, low viscosity, and high miscibility with other polymers. The physical and mechanical properties depend on its molecular weight and degree of crystallinity [9]. The PCL has a high degree of crystallinity (ranging from 30 to 60%), glass transition temperature of −60 °C and melting temperature (Tm) ranging between 50 to 60 °C [10].

Both polymers have characteristics that, when combined, tend to improve the performance of the polymer membrane as a biomaterial. One way of obtaining a new material with good properties of the individual components is through the preparation of polymer blends. Therefore, PLA/PCL blends were chosen because PLA presents biocompatibility, biodegradability, and good mechanical performance [11] while PCL has flexibility with a higher elongation at break, higher tensile strength, better permeability, and slower rate of enzymatic hydrolysis than PLA [12].

The membranes must present a high porosity, with an appropriate pore size, and interconnectivity of pores to provide proliferation, cell differentiation, and tissue growth [12]. Many techniques can be used to produce porous membranes, such as phase inversion [13], suspension polymerization reactions in particle form [14], phase separation and freeze drying, solution spinning, 3D printing [15], solvent casting with particulate leaching, gas foaming [16], and solvent casting in a controlled atmosphere [17]. The solvent casting in a controlled atmosphere method is easy to reproduce and has a low cost compared to other methods. In addition, it allows the production of pores with low variation in the distribution of size and homogeneous shapes. In order to obtain porous films, air humidity should be kept controlled at around 70–80%. As the consistency of the phases and the cooling process lead to the condensation of the water droplets, the porosity is formed by the surface tension at the polymer–water interface [17].

To improve the bioactivity properties, hydroxyapatite (HAp) was added to the PLA/PCL blend. The addition of hydroxyapatite may also improve the tensile strength of the blend, since 10–30 wt% of nanoapatite was able to increase the tensile strength of poly (lactic-co-glycolic acid) (PLGA) from 0.49 MPa to 0.61 MPa [18]. The polymer-ceramic composite is a promising alternative for guided bone regeneration since it presents excellent characteristics including mechanical resistance and flexibility, due to the use of the polymer matrix, and bioactivity, by the ceramic incorporation. HAp also increases the protein adsorption capacity, suppresses cell death by apoptosis, and creates a favorable microenvironment for bone regeneration [19].

In this context, the objective of this work was to produce and characterize the porous membranes of a PLA/PCL blend with the addition of HAp obtained by solvent casting in a controlled humidity method and evaluate the use of this material for guided bone regeneration.

## 2. Experiment

### 2.1. Materials

The poly (lactic acid) (PLA) was supplied by NatureWorks LLC with trade name Ingeo™Biopolymer 2003D and a density of 1.24 g/cm$^3$, and polycaprolactone (PCL) was supplied by Sigma-Aldrich, product number 440744 with a density of 1.145 g/cm$^3$ and Mw ~ 80,000 g/mol.

Hydroxyapatite (HAp) was produced in the laboratory using tetrahydrate calcium nitrate (Calcium Nitrate P.A.-A.C.S.-Labsynth®) and monobasic ammonium phosphate (Monobasic Ammonium Phosphate P.A. (Dinâmica®)) salts.

### 2.2. Synthesis of Hydroxyapatite (HAp)

Hydroxyapatite (HAp) was obtained by the precipitation method using a mixture of solutions of tetrahydrate calcium nitrate ($Ca(NO_3)_2 \cdot 4H_2O$) and monobasic ammonium phosphate ($NH_4H_2PO_4$) salts, with a molar ratio 5:3, both solubilized in deionized water. The mixture obtained was stirred until precipitation of HAp. The colloidal solution was dried in an oven at 50 °C for 24 h. The resultant powder of HAp was deagglomerated using a mortar and pestle [20–22].

### 2.3. Characterization of Hydroxyapatite (HAp)

The HAp was characterized by scanning electron microscopy (SEM) and X-ray diffraction (XRD). SEM measurements were performed using a field emission scanning electron microscope (MIRA3 TESCAN) and they were used to investigate the size and shape of the HAp powder. Phase crystalline was analyzed using an X-ray diffractometer (Philips, Amsterdam, The Netherlands, model X'Pert MRD), operating at 45 kV, 40 mA, and Cu–K$\alpha$ (1.540 Å). The analyses were performed using a grazing angle mode and the reflectivity spectrum was adjusted to scanning in $\frac{w}{2 \cdot \theta}$ between $w = 0.05°$ and $w = 7°$. The incident angle $w$ was $^1/_2$ of the detector angle $2\theta$.

### 2.4. Preparation of the Porous Membranes of PLA/PCL with Addition of Hydroxyapatite (HAp)

The HAp was dispersed in chloroform, under ultrasonic agitation. The ultrasound conditions used were potency of 200 W for 15 min and the temperature was kept under 40 °C. PLA and PCL in the ratio of 80/20 wt% were solubilized in a 10% *w/v* polymer/chloroform solution. The solubilization of the polymer was performed with constant mechanical agitation for 1 h at 40 °C. The resulting solution was dropped in molds with diameters of 9 mm and dried for 5 min at a controlled humidity (80%) by thermohygrometer (7666 model Incoterm) at room temperature. After that, the membranes were dried for 2 h in regular atmosphere [17].

PLA/PCL membranes with the addition of 5 wt% and 10 wt% of HAp were prepared and were named PLA/PCL-5% nHAp and PLA/PCL-10% nHAp. In addition, the PLA/PCL blend was named PLA/PCL.

### 2.5. Structural and Thermal Characterization of the Porous Membranes

The structural characterization of the porous membranes was evaluated using SEM. The same equipment and conditions described in the characterization of HAp particles were used.

The topography and roughness measurements were performed using an optical profilometer (WYKO NT 1100 series Optical Profiling System, Veeco, Plainview, NY, USA).

The thermal behavior was evaluated by differential scanning calorimetry (DSC) and thermogravimetric analysis (TGA). DSC analyses were performed using TA Instruments equipment (model QS 100). The samples were initially heated from room temperature to 200 °C at a heating rate of 10 °C/min. Then, the samples were cooled to −80 °C at a rate of 10 °C/min and afterwards reheated to 200 °C at the same heating ratio. The atmosphere was kept inert (nitrogen flow of 50 mL/min). The mass of the samples was kept constant at 10 mg. The degree of crystallinity (%C) was calculated using the following Equation (1):

$$\%C = \frac{\Delta H_m - \Delta H_{cc}}{\Delta H_{m0}} \tag{1}$$

where $\Delta H_{cc}$ is the cold crystallization enthalpy, $\Delta H_m$ is the melting enthalpy of the sample, and $\Delta H_{m0}$ is the melting enthalpy of a 100% crystalline sample. For the PLA, $\Delta H_{m0}$ is 93 J/g [7] and for PCL $\Delta H_{m0}$ is 136 J/g [23]. To obtain the degree of crystallinity, the Equation (1) was divided by a factor of the weight fraction of HAp.

Thermogravimetric analyses (TGA) were performed using TA Instruments equipment, model Q600, from 25 to 800 °C, at a heating rate of 20 °C/min. The mass of the samples was kept constant at 10 mg.

## 2.6. Biological Assays

### 2.6.1. Cell Culture

To assure reproducibility, the assays were repeated independently three times. MC3T3-E1 Subclone 14 cells (BCRJ: 0285) were obtained from Rio de Janeiro Cell Bank (Rio de Janeiro, Brazil). Cells were first cultured in growth medium ($\alpha$-MEM medium, supplemented with 10% fetal bovine serum-FBS and 1% antibiotic; Vitrocell, Campinas, Brazil) in a 75 $cm^2$ cell culture flask for 21 days. The samples were placed in an incubator at 37 °C with a humidified environment containing 5% $CO_2$. The medium was changed every 2–3 days. For all experiments, upon confluence, cells were then transferred to 24-well plates (10,000 cells per $cm^2$) and cultured for 3 days, 7 days, and 14 days with the compositions in an incubator in the same aforementioned conditions, with osteoinductive medium (growth medium with 1% β-Glycerophosphate, 1% 2-phospho-L-ascorbic acid trisodium salt, and 0.1% dexamethasone (Sigma-Aldrich, São Paulo, Brazil)) changed three times a week.

### 2.6.2. Cell Viability

The cytotoxicity of the membranes, the compositions, and their influence on cell viability were verified using an alamarBlue® assay (ThermoFisher Scientific, São Paulo, Brazil). MC3T3-E1 cells (BCRJ, RJ, Brazil) were cultured on the membranes for 3 days, 7 days, and 14 days. Afterwards, an alamarBlue® assay (Thermo Fisher Scientific, São Paulo, Brazil) was performed on all samples, at each time point (after 3 days, 7 days, and 14 days) in order to evaluate cell viability. For this analysis, each well was rinsed with phosphate buffered saline (PBS) to remove unattached cells and wash out the remaining serum, and 1 mL of 10% alamarBlue® solution (Thermo Fisher Scientific, São Paulo, Brazil) was added into each well and incubated in a dark humidified incubator set at 37 °C and with 5% $CO_2$ for 4 h. After that, 200 μL of solution (in duplicate) was aliquoted into wells of a 96-well plate for measurements at the spectrophotometric microplate reader (Bio-Tek Instruments Inc, Winooski, VT, USA) at 570 nm and 600 nm. From the values obtained, the MC3T3 cells viability rates were calculated using the percentage reduction of alamarBlue®, according to manufacturer's instructions.

After the alamarBlue® test, cells were washed away twice using MilliQ water, and the same well-plate at each experimental period was used for DNA quantification by PicoGreen assay (QuantiFluor® dsDNA quantification kit; Promega, São Paulo, Brazil). After three cycles of freeze–thaw (−20 °C and 25 °C), 100 μL of solution was added into each well, each well contained 100 μL of sample, and the plate was stored in the dark for 5 min. Finally, the fluorescent signal (485/20 excitation and 528/20 emission) was read using a microplate reader (Epoch Microplate Spectrophotometer, BioteK, Winooski, VT, USA). From the values obtained, the viability of MC3T3 cells can be determined.

## 3. Results and Discussion

### 3.1. Synthesis of Hidroxiapatite (HAp)

Figure 1 shows the SEM of the HAp nanoparticles obtained by the aqueous precipitation method. It is possible to observe particles of apparently needle shape and the formation of agglomerates of HAp. Different techniques report the creation of hydroxyapatite in nanometric sizes (nHAp), which contributes to its superficial reactivity, for example, Yubao et al. [24] obtained a crystal size of nHAp of 23 nm by 91 nm in the form of needles using the aqueous precipitation method, but using the drip technique of calcium nitrate in dibasic ammonium phosphate. Zhou et al. [25] reported obtaining nHAp with an average size of 80–90 nm by 15–30 nm via aqueous precipitation.

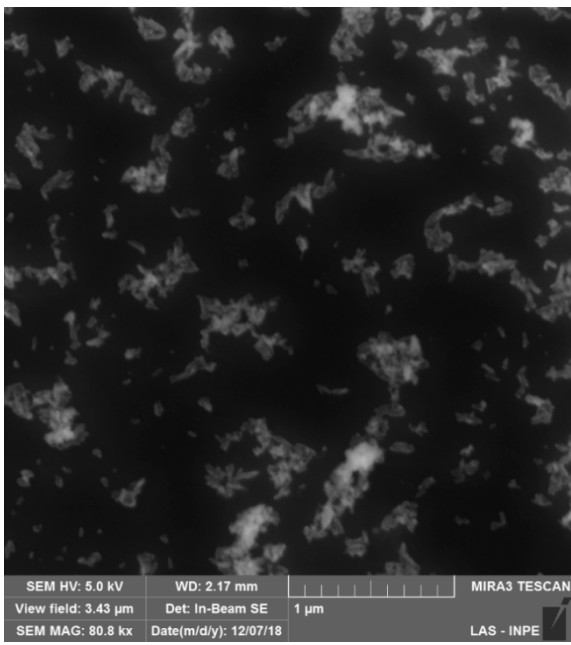

**Figure 1.** SEM image of hydroxyapatite (HAp) obtained by aqueous precipitation, 50,000× magnitude.

Figure 2 shows the XRD pattern of the HAp. Characteristic peaks are identified in the diffractogram and correspond to the peaks of the crystalline phase of HAp. The synthesis of calcium phosphates allows different products to be obtained by varying the Ca/P (calcium/phosphate) ratio of the reactants, so the characterization is necessary in order to prove the creation of HAp. Barbosa et al. [26] obtained diffraction peaks of nHAp in 2θ values of 25.9°, 31.9°, and 34.0°, also corresponding to HAp (file JCPDS 024-0033). Zhou et al. [25] obtained diffraction peaks in 2θ values of 25.9°, 31.9°, 40.1°, 47.1°, and 48.7°, corresponding to the file JCPDS 024-0033, which suggests that the nHAp retains the hexagonal structure. Singh et al. [27] obtained primary peaks attributed to HAp in values of 2θ of 25.9°, 32°, and 32.6°.

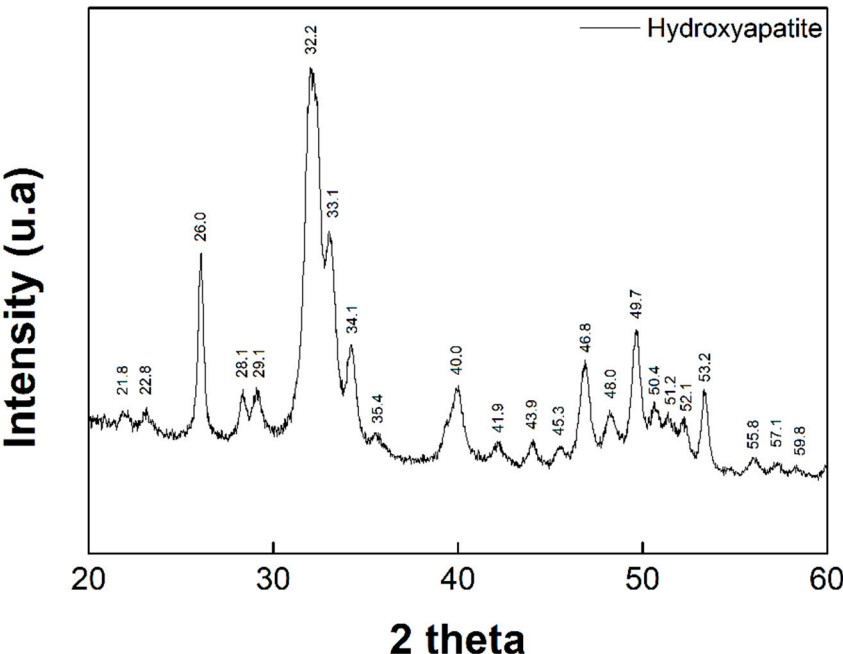

**Figure 2.** XRD of HAp obtained by aqueous precipitation.

*3.2. Structural and Thermal Characterization of Porous Membranes of PLA/PCL with Hydroxyapatite*

Figure 3 shows the SEM images of the PLA/PCL membranes (80/20 wt%) and the PLA/PCL blends-based composites with the of addition of 5 wt% and 10 wt% of HAp. The PLA/PCL membranes presented a medium pore size of 3.73 μm and 308 pores. The PLA/PCL-5% nHAp membrane had the highest content of pores (657) and, consequently, lower pore size (3.05 μm). The PLA/PCL-10% nHAp presented a higher content of pores than the PLA/PCL membranes (392) with larger diameters (4.79 μm).

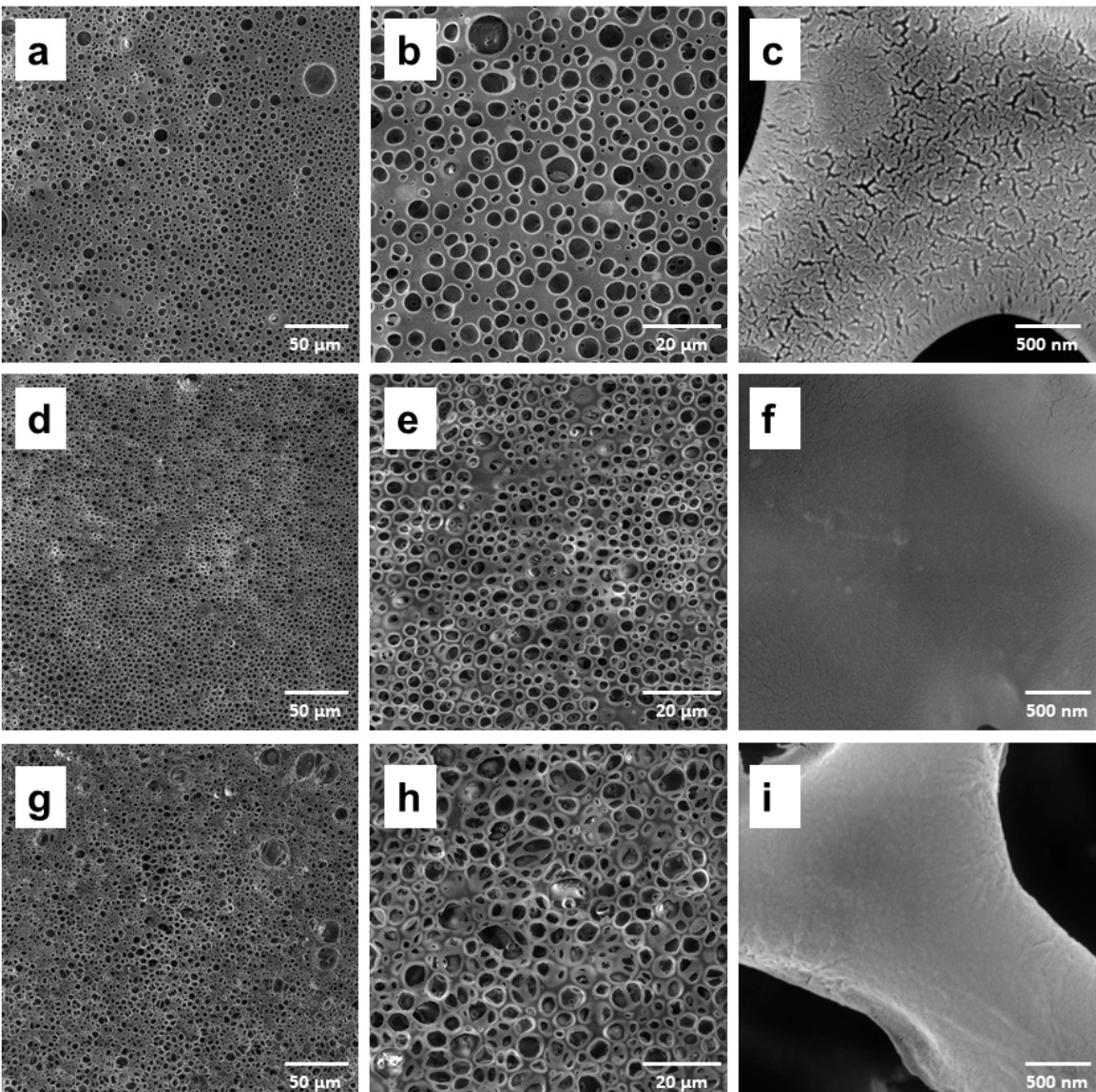

**Figure 3.** SEM images of porous membrane of poly (lactic acid) (PLA)/poly (ε-caprolactone) (PCL) (**a–c**), PLA/PCL-5% nHAp (**d–f**) and PLA/PCL-10% nHAp (**g–i**). In 1000× magnitude (**a,d,g**), in 3000× magnitude (**b,e,h**) and in 100,000× magnitude (**c,f,i**).

Analyzing the Figure 3c,f,i, it can be observed that the nanostructure of the blend without addition of HAp presents a fissured surface. Microcracks can provide greater fragility to the material and promote early degradation. This surface defect is minimized in the PLA/PCL blend-based HAp composites. The membrane containing 10 wt% of HAp presents the best surface properties, minimizing the original defects.

Figure 4 shows an increase in roughness on PLA/PCL blends-based HAp composite porous membranes (Figure 4b,c) compared to that of the PLA/PCL blend. These results prove that the preparation process in the incorporation of particles increased the surface roughness. Results of the roughness values are shown in Table 1.

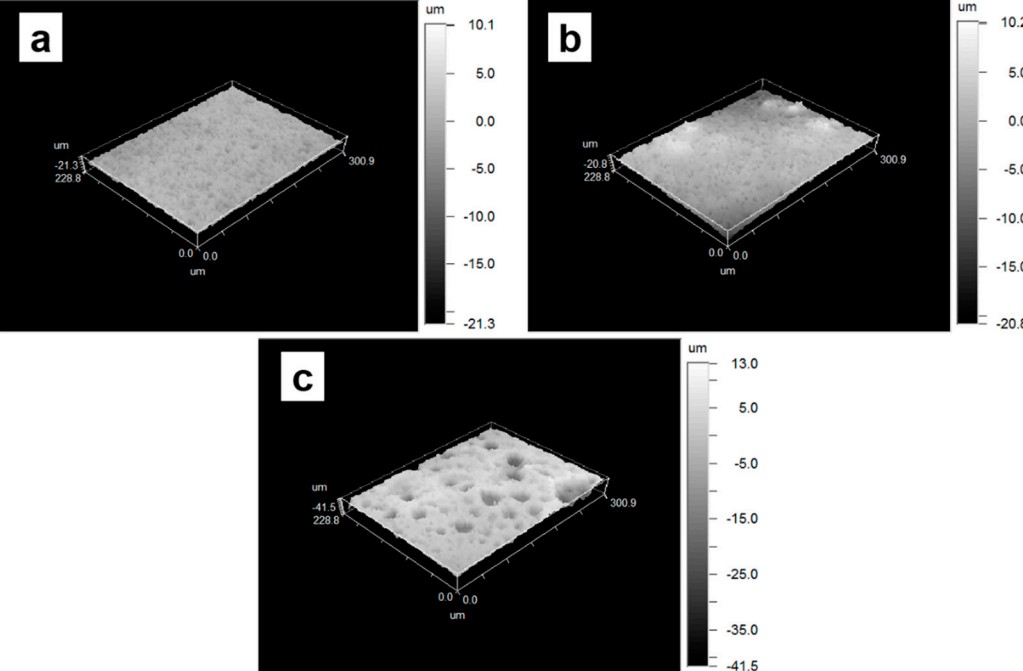

**Figure 4.** Profilometry of samples: (**a**) PLA/PCL, (**b**) PLA/PCL-5% nHAp, and (**c**) PLA/PCL-10% nHAp porous membranes.

**Table 1.** Mean surface roughness of the different samples.

| Samples | Surface Roughness (µm) |
|---|---|
| PLA/PCL | 1.24 ± 0.21 |
| PLA/PCL-5% nHAp | 1.74 ± 0.58 |
| PLA/PCL-10% nHAp | 3.92 ± 0.14 |

Figure 5 shows the XRD pattern of the polymer membranes. The characteristic bands of PLA and PCL, separately identified, are still present in the PLA/PCL polymer membrane in $2\theta$ values of 16.8°, 21.5°, and 23.9°. The characteristic bands of PCL have decreased intensity due to their low percentage present in the blend. It is also possible to observe peaks corresponding to the crystalline phase of HAp in $2\theta$ values of 26.0°, 28.1°, 29.1°, 32.2°, 33.1°, 34.1°, 40.0°, 43.9°, 46.8°, 48.0°, and 49.7°. The intensity of the peaks and their appearance depends on the content of HAp added to the polymer membrane. Similar observations were obtained in the characterization of PCL with HAp by Cardoso et al. [28].

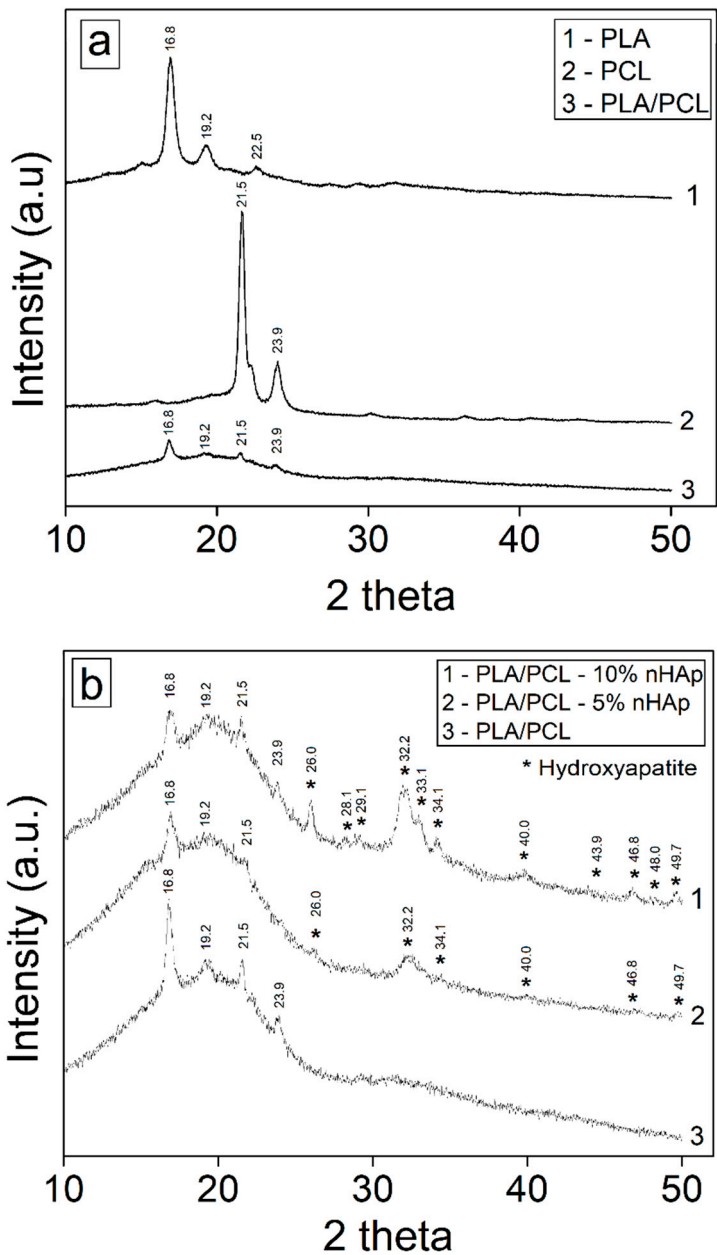

**Figure 5.** XRD of the porous polymer membranes of (**a**) PLA/PCL and (**b**) PLA/PCL, PLA/PCL-5% nHAp, and PLA/PCL-10% nHAp.

Figure 6 shows the DSC curves of the first heating of the PLA/PCL blends and the composites with different contents of HAp.

Table 2 presents data from the DSC curves of the neat PLA, neat PCL, PLA/PCL blend, and PLA/PCL blend-based composites with 5 wt% and 10 wt% of HAp for the first heating. The PLA/PCL blend is immiscible, and the individual temperatures and enthalpies of PLA and PCL were obtained. Neat PLA presents cold crystallization and this characteristic remained after blending with PCL. The addition of PCL decreased the glass transition temperature (Tg) and the degree of crystallinity (Xc) of the PLA in the blend when compared to those of the neat PLA. In an analogous way, the Xc of the PCL also decreased with the addition of PLA. It is possible to note that the decrease in the degree of crystallinity of the PCL phase was larger, with a decrease of about 49% in degree of crystallinity in the PLA/PCL blend when compared to that of the neat PCL. Analyzing the influence of the addition of HAp in the blends, a small decrease in Tg and Tm was observed in the PLA and PCL phases. These results

illustrate that the polymer–ceramic interaction can alter the mobility of the polymer chains, which directly affects its crystallinity [29]. The incorporated ceramic particles have a great influence in PLA and PCL. Several authors reported the change in the mobility of the polymer chains with the addition of particles, and as the degradation of a polymer is related to the degree of crystallinity of the polymer, the decrease in the crystallinity means the improvement in the rate of degradation of the membrane in biological systems [30].

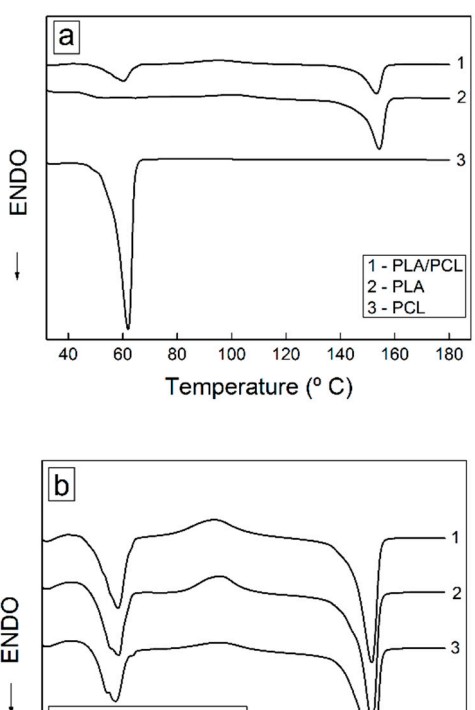

**Figure 6.** DSC curves of the first heating of (**a**) PLA, PCL, and PLA/PCL and (**b**) PLA/PCL, PLA/PCL-5% nHAp, and PLA/PCL-10% nHAp.

**Table 2.** DSC data of polymer membranes during the first heating.

| Sample | PLA | | | | | PCL | | |
|---|---|---|---|---|---|---|---|---|
| | $T_g$ (°C) | $T_{m1}$ (°C) | $\Delta H_{m1}$ (J/g) | $\Delta H_{cc1}$ (J/g) | $X_{c1}$ (%) | $T_{m1}$ (°C) | $\Delta H_{m1}$ (J/g) | $X_{c1}$ (%) |
| PCL | — | — | — | — | — | 56.0 | 77.4 | 57.0 |
| PLA | 57.8 | 148.4 | 24.7 | 3.0 | 23.4 | — | — | — |
| PLA/PCL | 53.0 | 146.0 | 19.2 | 3.4 | 16.9 | 52.0 | 11.0 | 8.1 |
| PLA/PCL-5% nHAp | 54.3 | 144.0 | 19.2 | 4.7 | 16.4 | 52.0 | 9.9 | 7.7 |
| PLA/PCL-10% nHAp | 54.6 | 146.0 | 17.8 | 1.5 | 19.4 | 52.0 | 6.0 | 4.9 |

Figure 7 and Table 3 show the TGA results of the porous membranes.

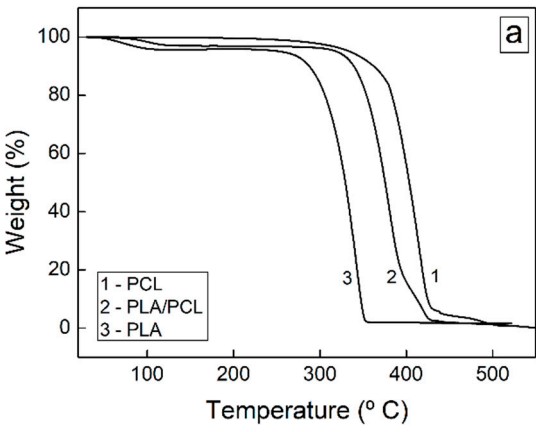

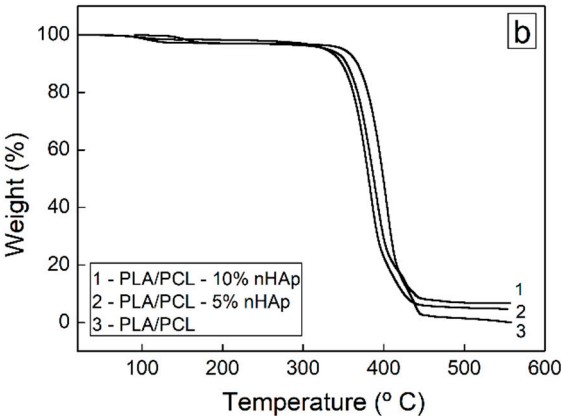

**Figure 7.** TGA of (**a**) PLA, PCL, and PLA/PCL and (**b**) PLA/PCL, PLA/PCL-5% nHAp, and PLA/PCL-10% nHAp.

**Table 3.** TGA data of the PLA/PCL blend and composites with different content of nHAp.

| Sample | $T_{onset\ of\ degradation}$ (°C) | Weight (%) | $T_{rreversible\ degradation}$ (°C) | Weigh Loss (%) | Residual (%) |
|---|---|---|---|---|---|
| PLA/PCL | 112.7 | 2.39 | 343.2 | 93.59 | 1.25 |
| PLA/PCL-5% nHAp | 79.4 | 1.36 | 357.6 | 92.02 | 5.08 |
| PLA/PCL-10% nHAp | 79.3 | 2.49 | 356.0 | 88.69 | 6.86 |

It is possible to observe two regions of mass loss in the curves. For the PLA/PCL blend, the first loss of mass occurs at 112.7 °C, which is only 2.39%, and may be related to evaporation of residual solvent from the membrane. The second mass loss occurs at 343.2 °C, which is the irreversible degradation temperature of the PLA/PCL blend, and the mass loss is 93.59%. The addition of HAp in the polymer matrix changes the degradation temperature. The first mass loss occurs at a lower temperature, about 79 °C. The second loss of mass occurs at a higher temperature, at 357 °C. The residual mass presents higher values than the residual mass of the PLA/PCL blend, and this is attributed to the homogeneous incorporation of inorganic filler to the polymeric matrix.

*3.3. Cell Viability*

Cell culture studies using alamarBlue® indicated that the porous polymeric membranes containing HAp (5 wt% and 10 wt%) did not affect MC3T3-E1 cell viability after 3 days, 7 days, and 14 days of culture, since no statistical difference was found when compared to PLA/PCL ($p > 0.05$) (Figure 8). DNA quantification by PicoGreen assay also indicated no significant difference among all the groups after 7 days and 14 days of culture ($p > 0.05$). On the other hand, at the first time point (3 days),

statistically lower values of DNA amounts were found for PLA/PCL-10% nHAp compared to PLA/PCL ($p$ = 0.0427) (Figure 9). No difference was observed among other groups in this period of time (3 days).

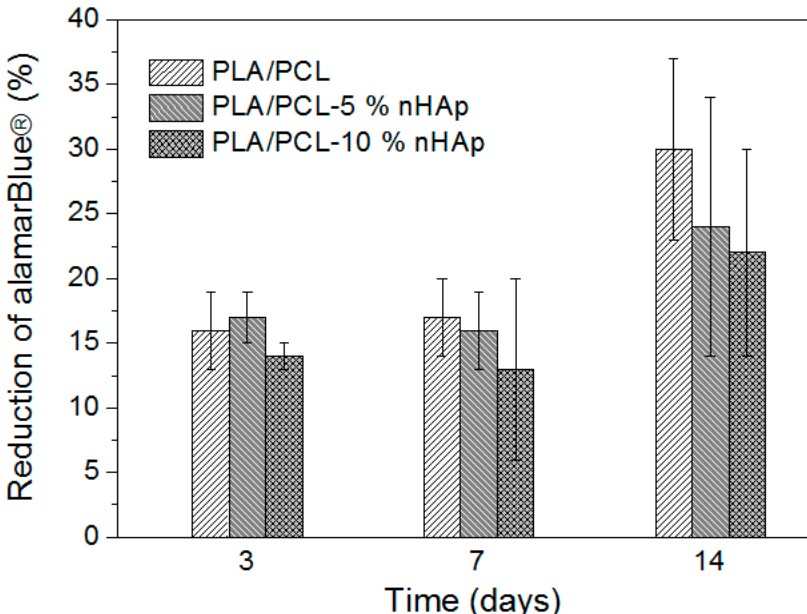

**Figure 8.** Graph of reduction of alamarBlue® for PLA/PCL, PLA/PCL with 5% *w/w* nHAp, and PLA/PCL with 10% *w/w* nHAp porous membranes in different experimental periods. Kruskal–Wallis test with Dunn's multiple comparison test.

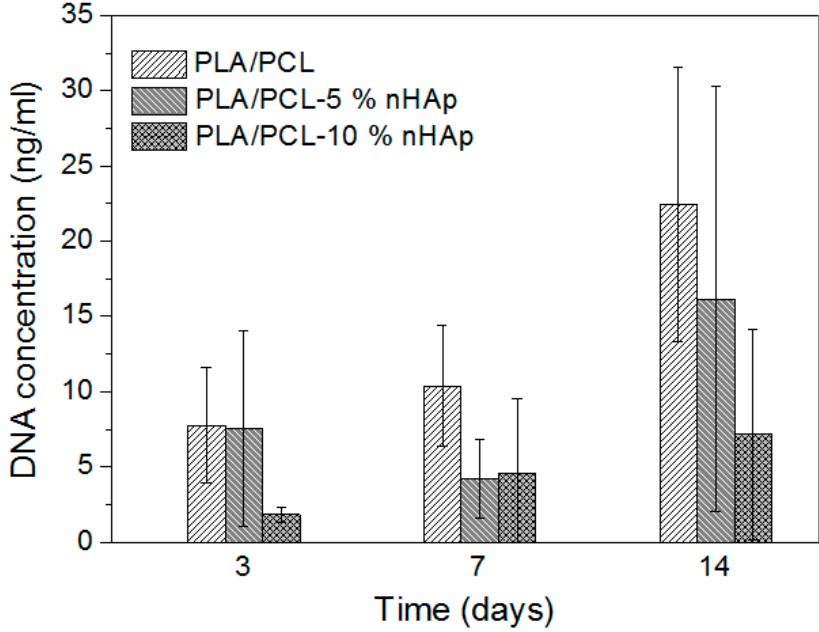

**Figure 9.** Graph of DNA concentration for PLA/PCL, PLA/PCL with 5% *w/w* nHAp, and PLA/PCL with 10% *w/w* nHAp membranes in different experimental periods. * PLA/PCL with 10% *w/w* nHAp compared to PLA/PCL ($p$ = 0.0427; Kruskal–Wallis test with Dunn's multiple comparison test).

In general, the results obtained in the cell viability studies showed that the incorporation of HAp (5 wt% and 10 wt%) into PLA/PCL membranes had no influence on cell viability, demonstrating that the HAp (in the different concentrations) was non-cytotoxic to MC3T3-E1 cells. Szymonoxicz et al. [31] using nHAp in powder form, obtained by the wet chemistry method at temperatures of 800 °C, 900 °C, and 1000 °C, also found that these materials show no cytotoxic effects on fibroblastic cells

(L929) [30]. Simionescu et al. [32] also verified that biocomposites obtained by combining functionalized nanoparticles of HAp with biopolymers and PCL were non-cytotoxic to human embryonic kidney 293 cells (HEK 293T cells) [32].

The incorporation of hydroxyapatite into a polymer matrix is already well described in the literature for its ability to biomimetize bone tissue [16,33]. The bone tissue is composed of collagen fibrils and an inorganic matrix composed of HAp nanocrystals [34,35], thus the polymers mimic the organic matrix and the HAp confers nucleation sites of new hydroxyapatite crystals for inorganic matrix formation. In addition, HAp also contributes to cell signaling mechanisms for bone matrix formation. The biomaterial developed in this work has a porous structure that confers greater cell adhesion and proliferation, these superficial properties are well described in the literature as favoring the regeneration of the tissue. The tests performed in this study evaluated the biomaterial interaction with the biological medium; the cells cultured on the surface showed good metabolic interaction with proliferation capacity on the surface of the material without cytotoxic effects. The results indicate the promising utilization of the PLA/PCL porous polymeric membranes (with 5 wt% and 10 wt% of HAp) for biomedical applications, since these composites are biocompatible and non-cytotoxic [36,37].

## 4. Conclusions

The aqueous precipitation technique for the synthesis of HAp resulted in a plate-like formed material, which is very promising for use in guided bone regeneration (GBR). A PLA/PCL blend and PLA/PCL blend-based HAp composites with different content of HAp were successfully produced using the controlled humidity solvent casting technique resulting in membranes of porous material. The addition of HAp provided an increased degradation temperature and roughness of the PLA/PCL blend. In addition, in cell culture studies, it was possible to observe that the membranes with HAp had no cytotoxic effect on MC3T3 cells. The incorporation of HAp in the PLA/PCL blend was satisfactory, with no alterations in precursor materials, thus obtaining a porous membrane that presents suitable properties for use in GBR.

**Author Contributions:** Conceptualization, E.d.S.T. and F.R.P.; Methodology, E.d.S.T., N.K.d.M., I.A.W.B.S. and F.R.P.; Validation, E.d.S.T., N.K.d.M., I.A.W.B.S. and F.R.P.; Formal Analysis, N.K.d.M., I.A.W.B.S., E.d.S.T., H.W.K., I.R.A., A.C.M.R., J.P.d.B.M. and F.R.P.; Investigation, N.K.d.M., I.A.W.B.S., E.d.S.T., H.W.K., I.R.A., A.C.M.R., J.P.d.B.M. and F.R.P.; Resources, E.d.S.T. and F.R.P.; Data Curation, E.d.S.T. and F.R.P.; Writing—Original Draft Preparation, N.K.d.M. and I.A.W.B.S.; Writing—Review & Editing, E.d.S.T. and F.R.P.; Visualization, N.K.d.M., I.A.W.B.S., E.d.S.T., H.W.K., I.R.A., A.C.M.R., J.P.d.B.M. and F.R.P.; Supervision, E.d.S.T. and F.R.P.; Project Administration, E.d.S.T. and F.R.P.; Funding Acquisition, E.d.S.T. and F.R.P.

**Funding:** This research was funded by Fundação de Amparo à Pesquisa do Estado de São Paulo (FAPESP) grant number 2015/24659-7 and 2016/19978-9 and Conselho Nacional de Desenvolvimento Científico e Tecnológico (CNPq).

**Conflicts of Interest:** The authors declare no conflict of interest.

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
