# Peer review of "Production and Characterization of Porous Polymeric Membranes of PLA/PCL Blends with the Addition of Hydroxyapatite"

_jcs, doi:10.3390/jcs3020045_

Round 1

Reviewer 1 Report

Dera Authors, I read with great interest your manuscript. 

I found it very interesting but before resubmission, in my opinion, it needs a revision for english and grammar style. Please, check it carefully. Check also the repetitions.

The manuscript must be well rewrite in order to better understand.

Further, same comments are:

Line 16 and Line 20: write the acronym explanation of PLA and PCL in  line 16 intend of line 20.

Line 25-26: keep off this sentence.

Line 45: change in ...... produces monomers and oligomers of lactic acid, ....

Line 47: the degree of crystallinity is not only 37% but it changes, like Tg and Tm.  Add further bibliographic references.

Line 52: report a medium value for PCL degree of crystallinity.

Line 71: what is PLGA?

Line 85: insert the acronym HAp

Line 89-91: adjust and rewrite the sentence.

Line 95: bracket after SEM and use the acronym HAp.

Line 96: do not repeat the explanation of the acronym SEM

Line 103: use the acronym HAp (control the whole text)

Line 113-115: it is a repetition. adjust the experimental part and write the same experimental procedure together.

Line 121: adjust the units. which is the sample weight?

Line 135-136: please, more explanation.

Line 138-139: adjust and rewrite the sentence.

Line 135-134: adjust all the paragraph

Line 149: why you choice this period of days (add bibliographic information)

Line 162: adjust the sentence 

Line 169: use only the acronym HAp.

Line 171-172: this information is superfluous.

Figure 2: it is not necessary to add the *. Redraw the figure

Figure 3: write the magnification in the figure or the the figure caption. Keep off the instrumental data in the bottom.

Table 1 cation: keep off N=3, it is not necessary.

Line 218: keep off " with different contents of nHAp (5 and 10% w/w)".

Table 2: adjust, keep off the first line and add the data of the first heating, that are the most important because are related to the original materials.

The experimental session, the materials and methods section as well as the conclusion session must be fully revised and improved.

Best regards.

Author Response

Response to Reviewer 1

The manuscript must be well rewrite in order to better understand.

R: We are so grateful for valuable comments about our study. All you contributions are been accepted.

Further, same comments are:

Line 16 and Line 20: write the acronym explanation of PLA and PCL in  line 16 intend of line 20.

R: The sentence was corrected

Line 25-26: keep off this sentence.

R: The sentence was corrected.

Line 45: change in ...... produces monomers and oligomers of lactic acid, ....

R: The sentence was corrected.

Line 47: the degree of crystallinity is not only 37% but it changes, like Tg and Tm.  Add further bibliographic references.

R: The sentence was corrected and new references were added.

Line 52: report a medium value for PCL degree of crystallinity.

R: Information has been added in the text.

Line 71: what is PLGA?

R: Information has been added in the text.

Line 85: insert the acronym HAp

R: Information has been added in the text.

Line 89-91: adjust and rewrite the sentence.

R: The sentence was rewritten

Line 95: bracket after SEM and use the acronym HAp.

R: The text was corrected.

Line 96: do not repeat the explanation of the acronym SEM

R: The text was corrected.

Line 103: use the acronym HAp (control the whole text)

R: The text was corrected.

Line 113-115: it is a repetition. adjust the experimental part and write the same experimental

procedure together.

R: The text was corrected.

Line 121: adjust the units. which is the sample weight?

R: Information has been added in the text.

Line 135-136: please, more explanation.

R: More information has been added in the text.

Line 138-139: adjust and rewrite the sentence.

R: The text was corrected.

Line 135-134: adjust all the paragraph

R: The text was corrected.

Line 149: why you choice this period of days (add bibliographic information)

R: The results presented are preliminary tests of cellular interaction, we present the first contact and response of cellular interaction with the biomaterial. The enzymatic assays at times of 3, 7, and 14 days are enough to prove the adhesion and survival capacity of the cells in contact with the biomaterial. Cell adhesion is the first phase that will influence the ability of cells to proliferate

and differentiate in contact with the implant. Enzyme cell viability tests are determined at 14 days and monitored at shorter intervals to check for possible changes in metabolism and cell death. Similar methodologies can be seen in https://doi.org/10.1039/C8BM00693H and https://doi.org /10.1186/s13036-017-0074-3.

Line 162: adjust the sentence 

R: The text was corrected.

Line 169: use only the acronym HAp.

R: The text was corrected.

Line 171-172: this information is superfluous.

R: The text was corrected.

Figure 2: it is not necessary to add the *. Redraw the figure

R: The Figure was changed.

Figure 3: write the magnification in the figure or the the figure caption. Keep off the instrumental data in the bottom.

R: The text was corrected and the Figure was changed.

Table 1 cation: keep off N=3, it is not necessary.

R: The text was corrected.

Line 218: keep off " with different contents of nHAp (5 and 10% w/w)".

R: The text was corrected.

Table 2: adjust, keep off the first line and add the data of the first heating, that are the most important because are related to the original materials.

R: We agree with the suggestion. To answer and leave the information clearer we replace the data of the second heating by the data of the first heating, including graphs and table.

The experimental session, the materials and methods section as well as the conclusion session

just be fully revised and improved.

R: All sections have been reviewed and improved.

Reviewer 2 Report

The authors submitted the article entitled “Production and characterization of porous polymeric membranes of PLA/PCL blends with the addition of hydroxyapatite  ”. I recommend that the paper could be accepted after minor revisions. My main comments and questions are as follows:

1. The authors should add mechanism or process for generating porous films.

2. Can the authors control the pore size?

3. How about the difference of mechanical properties since it’s important for applications.

4. Numbering should be utilized in the figures of XRD, DSC, and TGA since they were unclear.

5. In Fig 9, the authors should add the reference of flat films with and without nHAp to let reader comprehend the important factors.

6. The authors should check the format of the references.

7. English correction is recommended.

Author Response

Response to Reviewer 2

R: We are so grateful for valuable comments about our study. All you contributions are been accepted.

The authors submitted the article entitled “Production and characterization of porous polymeric membranes of PLA/PCL blends with the addition of hydroxyapatite  ”. I recommend that the paper could be accepted after minor revisions. My main comments and questions are as follows:

1.      The authors should add mechanism or process for generating porous films.

R: The text with the process explanation was added.

2.      Can the authors control the pore size?

R: To control the pore size you control the humidity and time that the solution stay under the atmosphere. The humidity (80%) was controlled by thermohygrometer (7666 model, Incoterm), this information has been added in the text.

3.      How about the difference of mechanical properties since it’s important for applications.

R: We agree that changes in mechanical properties are very important. The study group is developing a methodology to verify the mechanical properties and future trials will be carried out.

4.      Numbering should be utilized in the figures of XRD, DSC, and TGA since they were unclear.

R: We agree with the suggestion. The figures were changed.

5.      In Fig 9, the authors should add the reference of flat films with and without nHAp to let reader comprehend the important factors.

R: We added a reference in the sentence.https://www.ncbi.nlm.nih.gov/pmc/articles/PMC5641988/

6.      The authors should check the format of the references.

R: The references were checked and corrected.

7.      English correction is recommended.

R: An extensive review was performed throughout the manuscript.

Round 2

Reviewer 1 Report

Dear Authors, 

in my opinion, the manuscript was improved and it is susceptible to be accepted of republication.

With my best regards

Author Response

We would like to acknowledge the acceptance of our manuscript.

Reviewer 2 Report

The paper has revised well.

Author Response

(The authors gave the same response as above.)
